# ME3BP-7 is a targeted cytotoxic agent that rapidly kills pancreatic cancer cells expressing high levels of monocarboxylate transporter MCT1

Jordina Rincon-Torroella[1,2,3†], Marco Dal Molin[4†], Brian Mog[1,5,6,7], Gyuri Han[1], Evangeline Watson[1], Nicolas Wyhs[1,3,6,8], Shun Ishiyama[9], Taha Ahmedna[9], Il Minn[10], Nilofer Azad[3], Chetan Bettegowda[1,2,11], Nickolas Papadopoulos[1,3,6,11], Kenneth W Kinzler[1,3,6,8,11], Shibin Zhou[1,3,6,11], Bert Vogelstein[1,3,5,6,8,11]*, Kathleen Gabrielson[3,8,9]*, Surojit Sur[1,3,6,11]*

[1]Ludwig Center, Sidney Kimmel Comprehensive Cancer Center, The Johns Hopkins University School of Medicine, Baltimore, United States; [2]Department of Neurosurgery, The Johns Hopkins University School of Medicine, Baltimore, United States; [3]Department of Oncology, The Johns Hopkins University School of Medicine, Baltimore, United States; [4]Department of Surgery, The Johns Hopkins University School of Medicine, Baltimore, United States; [5]Howard Hughes Medical Institute, Chevy Chase, United States; [6]Lustgarten Pancreatic Cancer Research Laboratory, Sidney Kimmel Comprehensive Cancer Center, The Johns Hopkins University School of Medicine, Baltimore, United States; [7]Department of Biomedical Engineering, Johns Hopkins University, Baltimore, United States; [8]Department of Pathology, The Johns Hopkins University School of Medicine, Baltimore, United States; [9]Department of Molecular and Comparative Pathobiology, The Johns Hopkins University School of Medicine; Sidney Kimmel Comprehensive Cancer Center, Johns Hopkins University, Baltimore, United States; [10]Division of Nuclear Medicine and Molecular Imaging, The Russell H. Morgan Department of Radiology and Radiological Science, The Johns Hopkins University School of Medicine, Baltimore, United States; [11]Bloomberg~Kimmel Institute for Cancer Immunotherapy, Sidney Kimmel Comprehensive Cancer Center, Baltimore, United States

*For correspondence:
vogelbe@jhmi.edu (BV);
kgabriel@jhmi.edu (KG);
ssur1@jhmi.edu (SS)

†These authors contributed equally to this work

## eLife assessment

This study presents a **valuable** finding and developed ME3BP-7 as a novel microencapsulated formulation of 3BP, which specifically targets MCT1-overexpressing PDAC cells. It demonstrates its specificity and efficacy in vitro and in PDAC mouse models, with significant anti-tumor effects and improved serum stability. Overall, the evidence supporting the authors' claims is **solid**.

**Abstract** Nearly 30% of pancreatic ductal adenocarcinomas (PDACs) exhibit a marked over-expression of monocarboxylate transporter 1 (MCT1) offering a unique opportunity for therapy. However, biochemical inhibitors of MCT1 have proven unsuccessful in clinical trials. In this study, we present an alternative approach using 3-bromopyruvate (3BP) to target MCT1 overexpressing PDACs. 3BP is a cytotoxic agent that is known to be transported into cells via MCT1, but its clinical usefulness has been hampered by difficulties in delivering the drug systemically. We describe here a

novel microencapsulated formulation of 3BP (ME3BP-7), which is effective against a variety of PDAC cells in vitro and remains stable in serum. Furthermore, systemically administered ME3BP-7 significantly reduces pancreatic cancer growth and metastatic spread in multiple orthotopic models of pancreatic cancer with manageable toxicity. ME3BP-7 is, therefore, a prototype of a promising new drug, in which the targeting moiety and the cytotoxic moiety are both contained within the same single small molecule.

## Introduction

Pancreatic ductal adenocarcinoma (PDAC) confers a dismal 5-year survival (*Sant et al., 2009*; *Jemal et al., 2009*). Even after aggressive multimodal treatment, only a minority of patients achieve long-term survival. The mainstay of treatment remains combinations of fluorouracil, irinotecan, oxaliplatin (FOLFIRINOX), or gemcitabine/nab-paclitaxel, which cause severe side effects and offer only modest improvements in survival (*Conroy et al., 2011*; *Conroy et al., 2018*; *Von Hoff et al., 2013*). Therefore, the development of new rationally designed therapeutic agents is critical for improved outcomes in patients diagnosed with this devastating disease.

Metabolic reprogramming is one of the hallmarks of PDACs. Major effectors underlying the aberrant metabolic re-wiring in cancer are members of the monocarboxylate transporter (MCT) family. These transmembrane proteins mediate the transport of pyruvate, lactate, short-chain fatty acids, and ketones in and out of cells (*Halestrap and Meredith, 2004*; *Halestrap, 2012*; *Halestrap and Wilson, 2012*). Several studies have associated increased monocarboxylate transporter 1 (MCT1) expression in various cancers with chemoresistance and poor prognosis (*Sant et al., 2009*; *Leu et al., 2021*; *Chen et al., 2010*; *Chen et al., 2019a*; *Simões-Sousa et al., 2016*; *Pinheiro et al., 2008*; *Miranda-Gonçalves et al., 2021*; *Latif et al., 2017*; *Sandforth et al., 2020*). Although MCT1 is an intriguing biological target, the small molecule MCT1 inhibitor AZD3965 was not found to be effective in the treatment of solid tumors (*Beloueche-Babari et al., 2020*; *Benyahia et al., 2021*).

3-Bromopyruvate (3BP) is a potent cytotoxic analog of pyruvic acid with a unique mode of action (*Darabedian et al., 2018*). The anticancer effects of 3BP were initially attributed to inhibition of glycolysis (*Ganapathy-Kanniappan et al., 2009*; *Tang et al., 2012*; *Cardaci et al., 2012*). However, recent studies have revealed 3BP as an alkylating agent of multiple intracellular proteins (*Darabedian et al., 2018*; *Ko et al., 2001*). MCT1 expression is essential for the activity of 3BP (*Birsoy et al., 2013*), although effective methods to systemically deliver 3BP have been a major research challenge. Clinical development of 3BP has been hampered by its poor serum stability, unfavorable pharmacokinetics, and excessive in vivo toxicity despite substantial preclinical and clinical efforts (*Azevedo-Silva et al., 2016*; *Chang et al., 2007*). The biochemical basis for these poor pharmacokinetic features is that 3BP alkylates free sulfhydryl groups in plasma proteins, including albumin, resulting in drug inactivation (*Darabedian et al., 2018*). To overcome this rapid loss of 3BP activity, large doses of the drug have been administered systemically, or it was delivered locally to tumors, such as through the hepatic artery in the case of liver tumors (*Vali et al., 2008*).

Our previous work demonstrated that microencapsulation of 3BP reduces its toxicity (*Chapiro et al., 2014*). Herein, we describe a novel encapsulated formulation of 3BP, ME3BP7, that reduces its degradation in serum. We show that ME3BP7 preferentially causes cell death in cells expressing MCT1 and inhibits the growth of orthotopic PDAC xenografts in mice without excessive toxicity.

## Results

### MCT1 expression mediates sensitivity of PDAC cell lines to 3BP

Comparison of RNAseq datasets from TCGA and the Cancer Cell Line Encyclopedia (CCLE) with The Genotype-Tissue Expression (GTEx) portal revealed that ~20–25% of PDACs exhibit increased expression of *SLC16A1*, the gene encoding MCT1, a mediator of 3BP activity. We, then, explored the activity of 3BP as a free drug in a panel of PDAC cell lines, including MIA PaCa-2, PSN-1, Panc 02.13, AsPC-1, BxPC-3, and CFPAC-1. Cells were treated with increasing concentrations of 3BP (0–220 μM) and assessed for cell death with real-time quantitative live-cell imaging. Five of the cell lines, MIA PaCa-2, PSN-1, Panc 02.13, AsPC-1, and BxPC-3, were sensitive to 3BP, with IC50s ranging from 24 to 40 μM (*Figure 1A and B*). In contrast, CFPAC-1 was highly resistant to 3BP, at concentrations of up

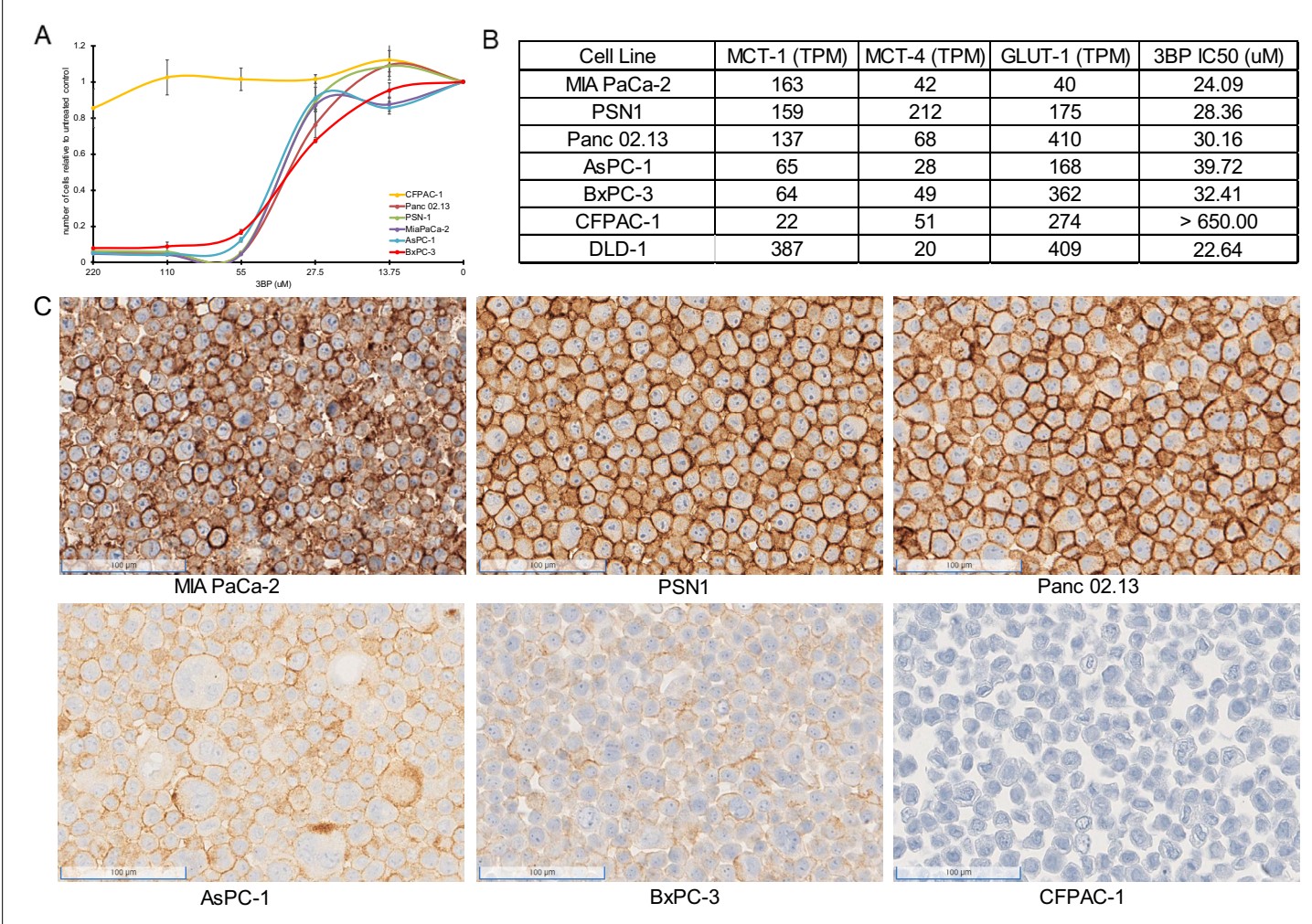

**Figure 1.** 3-Bromopyruvate (3BP) sensitivity of pancreatic ductal adenocarcinoma (PDAC) cell lines. (**A**) Response of six different PDAC cell lines to 3BP. The indicated cell lines were exposed to increasing doses of 3BP for 72 hr and evaluated with a SYBR green growth assay. Data are represented as the mean ± SD of three technical replicates and are normalized to untreated controls. (**B**) Expression levels of *MCT1, MCT4,* and *GLUT1* (TPM) and corresponding IC50s of 3BP. (**C**) Immunohistochemistry performed on PDAC cell lines with monoclonal mouse antibody against monocarboxylate transporter 1 (MCT1).

to 220 µM. RNA expression levels of three cellular transporters (*Cunningham et al., 2022*; *Barretina et al., 2012*), *SLC2A1* (GLUT1), *SLC16A1* (MCT1), and *SLC16A3* (MCT4), were examined to identify potential associations between the drug and cell viability (*Figure 1B*). The expression of *MCT1* was much lower in the 3BP-resistant cell line CFPAC-1 than in the other five cell lines. However, no clear relationship was found between 3BP resistance and the expression of the other transporters. MCT1 immunohistochemical (IHC) analyses furthermore demonstrated that the expression of the MCT1 protein on the cell surface paralleled RNA levels in these cell lines (*Figure 1C*).

To further demonstrate the specificity of the drug, we genetically inactivated MCT1 in cells and compared the viability of inactivated cells with their parental counterparts in the presence of 3BP. A similar strategy was used to establish MCT1 as essential for 3BP-mediated cell death in KBM7 cells, a unique near-haploid cell line derived from a leukemia (*Birsoy et al., 2013*). We deleted the gene encoding MCT1, *SLC16A1*, in the 3BP-sensitive cell line MIA PaCa-2 that expresses high levels of MCT1 (*Figure 1A and B*). Next-generation sequencing confirmed knockout of MCT1 in six clones (*Figure 2A and B*), and the results were validated with IHC (*Figure 2C and D*). Cells from the six clones were then mixed at equal ratios to create an MCT1 KO cell population (KO cells), thereby avoiding potential variables often associated with the evaluation of a single clone (*Torrance et al., 2001*; *Lee et al., 2019*).

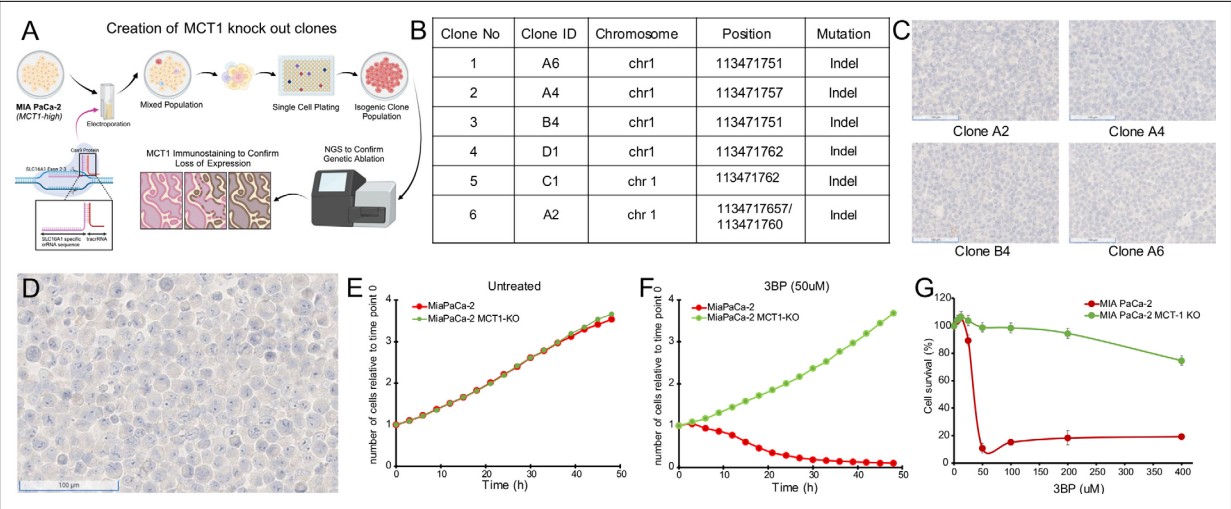

**Figure 2.** Monocarboxylate transporter 1 (MCT1) mediates 3-bromopyruvate (3BP) activity. (**A**) Strategy for CRISPR-based knockout of *SLC16A1* in MIA PaCa-2 cells. Created using BioRender.com. (**B**) Table of knockout clones. (**C**) Immunohistochemical analyses performed on representative KO clones with monoclonal mouse antibody against MCT1 (1:2000 dilution). (**D**) IHC performed with MCT1 antibody on pooled monoclonal MCT1 KO cells used to test the MCT1-specific activity of 3BP and ME3BP-7. (**E, F**) Comparison of cell growth over time of MIA PaCa-2 and MIA PaCa-2 MCT1 KO in the absence and presence of 3BP (50 µM) normalized to time point 0 hr. Data are represented as the mean ± SD of two technical replicates. (**G**) Dose-response curves of MIA PaCa-2 cells and MIA PaCa-2 MCT1 KO at 36 hr. Cell viability normalized to the number of cells at 0 hr. Data represent the mean ± SD of two technical replicates.

Next, we introduced nuclear-restricted green and red fluorescent proteins into the KO and parental cells, respectively, to enable simultaneous visualization in co-culture. KO cells grew at a rate indistinguishable from the parental cells under normal culture conditions (*Figure 2E*). However, in the presence of 50 µM 3BP, the parental cells (*Figure 2G*), were almost completely eliminated (IC50 24 uM) while the KO cells continued to proliferate (*Figure 2F* and *Videos 1 and 2*). These results indicated the specificity of 3BP for cells expressing MCT1.

## Optimal encapsulation of 3BP protects from degradation by human serum

One of the major challenges of using 3BP as a chemotherapeutic agent is that it is rapidly inactivated by the serum proteins it alkylates before it leaves the circulation and enters cancer cells (*Chang et al., 2007*). We have previously shown that encapsulation of 3BP with a cyclodextrin can mitigate its toxicity, presumably by permitting the use of lower concentrations of the encapsulated

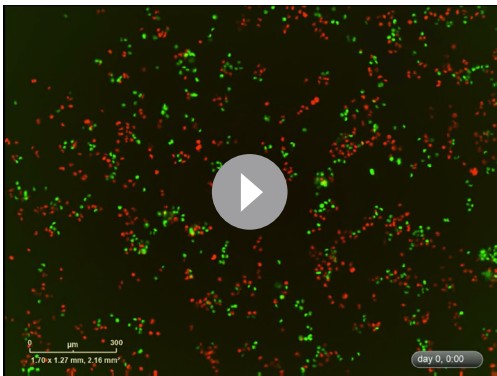

**Video 1.** Time-lapse movies of specific killing of monocarboxylate transporter 1 (MCT1)-expressing cells, untreated controls.
https://elifesciences.org/articles/94488/figures#video1

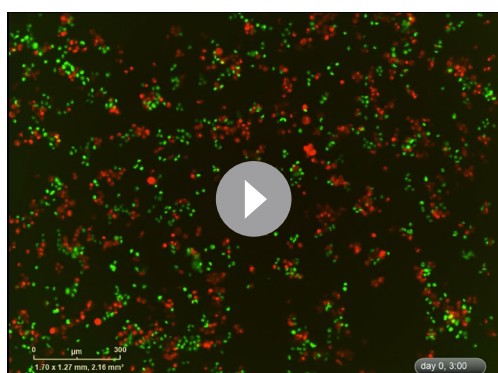

**Video 2.** Time-lapse movies of specific killing of monocarboxylate transporter 1 (MCT1)-expressing cells with 3-Bromopyruvate (3BP).
https://elifesciences.org/articles/94488/figures#video2

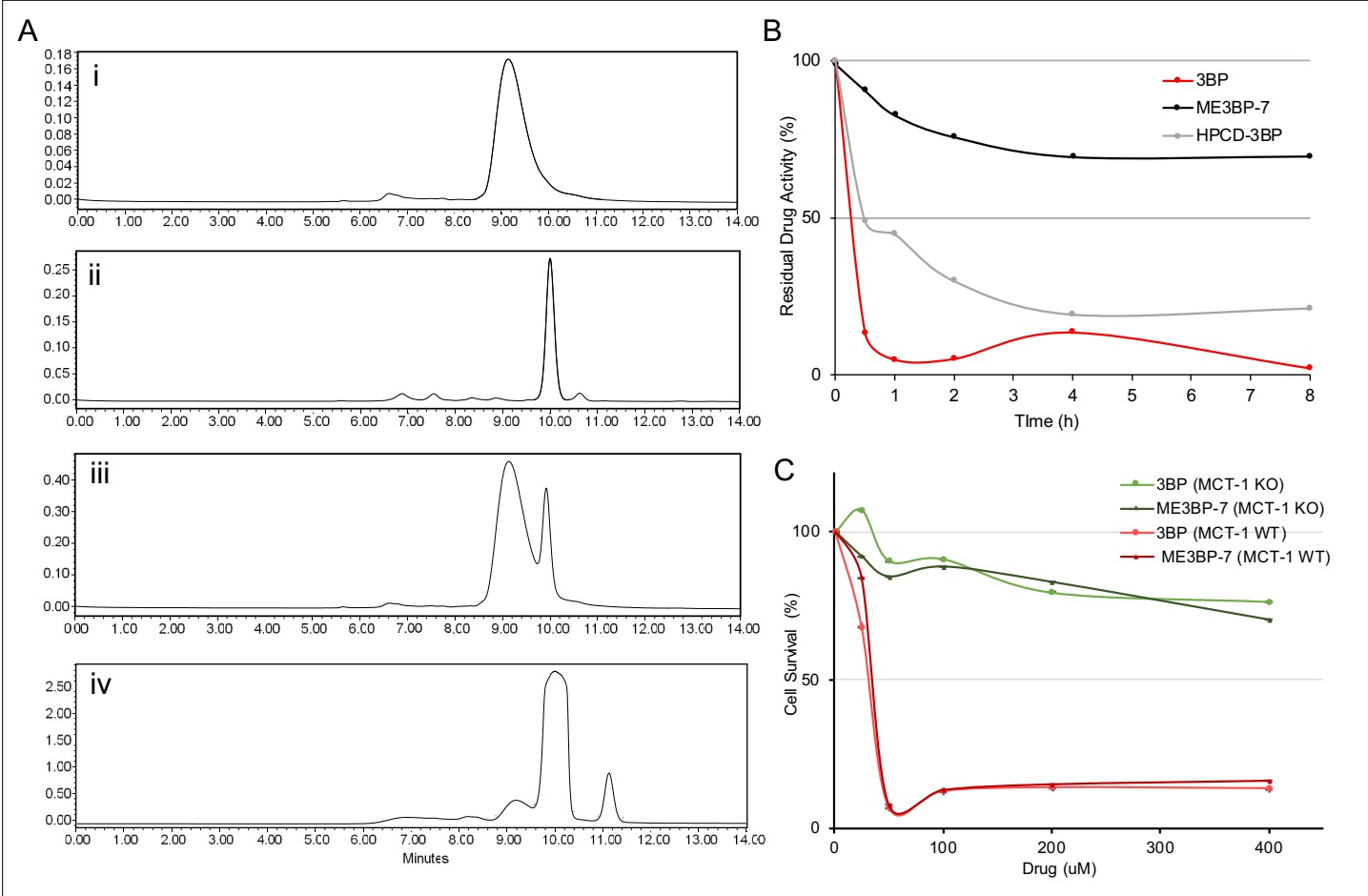

**Figure 3.** New formulations and serum stability of 3-bromopyruvate (3BP) in cyclodextrin complexes. (**A**) HPLC: Evaluation of different microencapsulated β-cyclodextrin complexes using size-exclusion chromatography (SEC). The agents examined were (i) 3BP (1 mg/mL), (ii) succinyl-β-CD (20 mg/mL), (iii) a mixture of 10 µL of 3BP and 10 µL succinyl-β-CD, and (iv) ME3BP-7 (10 mg/mL). Samples were monitored at 220 nm as shown in A–D. (**B**) Serum stability assay using DLD-1 cells. (**C**) ME3BP-7 specificity assessed with MIA PaCa-2 parental and MIA PaCa-2 monocarboxylate transporter 1 (MCT1) KO cells.

drug compared to the free drug (*Chapiro et al., 2014*). Therefore, we focused on optimizing the microencapsulation procedure and then examined the serum stability of the drug. To optimize microencapsulation of 3BP in cyclodextrins, we chose the β-cyclodextrin scaffold rather than alpha or gamma cyclodextrins based on structural modeling (*Chapiro et al., 2014*). We first evaluated two types of β-cyclodextrin modifications, one with the hydroxyls substituted with succinyl groups and the other with the hydroxyls substituted with 2-hydroxypropyl groups. Second, for the modifications involving succinyl groups, we sought to determine the optimal number of hydroxyl substitutions. Finally, we investigated the optimal ratio of cyclodextrin to 3BP. To evaluate these various formulations, we first used size-exclusion chromatography (SEC) to ensure that the amount of free 3BP was minimized (*Figure 3A*). Second, we used cell toxicity as a measure of the stability of these formulations. Because the cytotoxicity of 3BP decreases upon interaction with serum proteins, residual cell toxicity following exposure to serum is directly related to its resistance to serum degradation. Therefore, the formulations were incubated with human serum at 37°C and aliquots were collected at various time points for assessment of cell toxicity in DLD-1 cells which have high expression of MCT1 and sensitivity to 3BP (IC50, 22.64 µM; *Figure 1B*). Among the β-cyclodextrin formulations tested, we found that succinyl-substituted cyclodextrins were capable of protecting 3BP in sera more efficiently than hydroxypropyl-substituted versions. Moreover, a succinyl-substituted cyclodextrin with an average of 3.4 succinyl groups per cyclodextrin, at a molar ratio of 1.2 β-cyclodextrin per 3BP, performed best (*Figure 3A*). This MicroEncapsulated

formulation was named 'ME3BP-7'. Although free 3BP lost 90% of its activity by 30 min of exposure to serum, ME3BP-7 lost <10% of its activity in 30 min (*Figure 3B*). Remarkably, even after 8 hr of incubation with serum, ME3BP-7 retained >70% of its activity (*Figure 3B*). Hydroxypropyl-substituted cyclodextrin-3BP complexes were more stable than free 3BP, losing just over 50% of its activity after 30 min of exposure to serum. However, these complexes were not nearly as stable as ME3BP-7 (*Figure 3B*).

Finally, parental MIA PaCa-2 (MCT1 WT) in contrast to MCT1 KO cells were sensitive to the encapsulation of 3BP within ME3BP-7, indicating that the compound retained specificity as well as cytotoxicity for cells expressing MCT1 (*Figure 3C*).

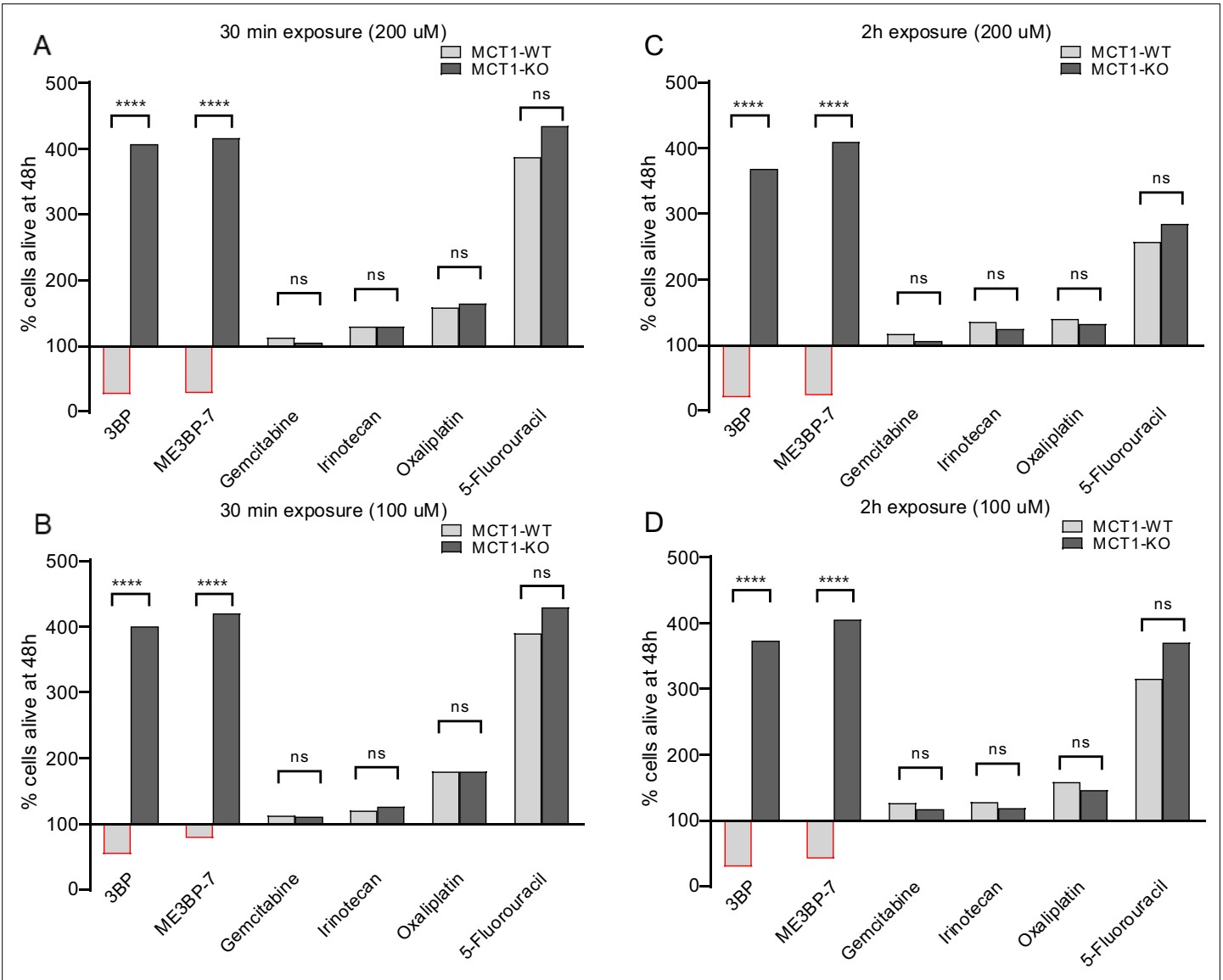

**Figure 4.** Comparison of monocarboxylate transporter 1 (MCT1)-specific cytotoxicity of 3-bromopyruvate (3BP), ME3BP-7, and current standard of care agents for pancreatic ductal adenocarcinoma (PDAC) upon short exposures. Viability of MCT-1 isogenic panel after (**A**) drug exposure for 30 min at 200 μM. (**B**) Drug exposure for 30 min at 100 μM. (**C**) Drug exposure for 2 hr at 200 μM. (**D**) Drug exposure for 2 hr at 100 μM.

The online version of this article includes the following figure supplement(s) for figure 4:

**Figure supplement 1.** Incucyte data for 15 min exposure for higher doses of various drugs.

**Figure supplement 2.** Linear scatter plots comparing the growth (or death) of MIA PaCa-2 parental after short exposure to 3-bromopyruvate (3BP), ME3BP-7 (reds), and current standard pancreatic ductal adenocarcinoma (PDAC) chemotherapeutic agents (grays).

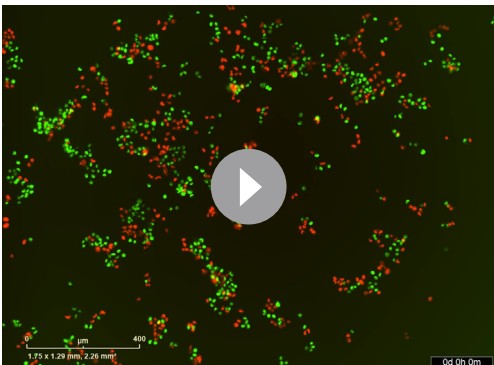

**Video 3.** Time-lapse movie of monocarboxylate transporter (MCT) WT vs KO exposed to oxaliplatin (200 µM) for 2 hr.

https://elifesciences.org/articles/94488/figures#video3

## ME3BP-7 results in rapid target-cell killing

We also noticed that the morphology of the MIA PaCa-2 cells rapidly changed in response to exposure to 3BP or ME3BP-7. To determine the significance of this change in morphology, we performed washout experiments on co-cultures of WT and KO cells exposed to drugs. Co-cultures were exposed to 3BP or ME3BP-7 at various concentrations and time periods, and cell proliferation was followed for 48 hr after removal of the drug. Exposure to either 3BP or ME3BP-7 for as little as 30 min led to the loss of the majority of WT MIA PaCa-2 cells. In contrast, KO MIA PaCa-2 cells continued to proliferate under all concentrations of the drugs and time points tested (*Figure 4A–D*, *Figure 4—figure supplements 1 and 2*). In addition, gemcitabine or any of the FOLFIRINOX agents, including irinotecan, oxaliplatin (*Video 3*), and 5-FU, exhibited no differential effect on WT and KO cells, thus emphasizing the specificity of 3BP and ME3BP-7 for MCT1-expressing cells. Moreover, 3BP and ME3BP-7 caused cell death of MIA PaCa-2 cells, while the standard of care agents only decreased their proliferation (*Figure 4*, *Figure 4—figure supplements 1 and 2*).

## ME3BP-7 shows efficacy in vivo

To assess efficacy in vivo, we treated mouse models of PDAC with ME3BP-7. We first tested the feasibility of delivering ME3BP-7 systemically and compared the toxicity of free 3BP with that of ME3BP-7 in athymic nude mice. Escalating doses ranging from 8.0 to 33 mg/kg of free 3BP (as ME3BP-7) were used to identify a maximum tolerated dose of ME3BP-7. Equivalent amounts of free 3BP and ME3BP-7 were infused via vascular access buttons (VABs) (33 mg/kg of 3BP) in 200 µL of PBS every other day for 1 week. In preliminary experiments, two-thirds of mice in the free 3BP group became ill, showing irrecoverable weight loss, pallor, and decreased activity, and died after only two doses. No overt toxicity was evident even after 5 weeks of treatment with ME3BP-7 at the same dose (33 mg/kg) (*Figure 5—figure supplement 1A*).

We then generated orthotopic xenografts with Panc 02.13, a patient-derived PDAC cell line with high expression of MCT1 (RNA expression of 137 transcripts per million) (*Figure 1B*), engineered to express firefly luciferase. Tumor-bearing animals were randomized into three treatment arms on the basis of intravital multiphoton imaging (IVIS), and treatment was initiated a day later (*Figure 5A*). Guided by our dose-finding toxicity studies, no treatment arm with free 3BP was included in this study as none of the animals would survive the course of the therapy. Over the course of 4 weeks of treatment (dosed Mondays, Wednesdays, and Fridays), no discernible adverse effects or weight loss was observed in the treated animals (*Figure 5—figure supplement 1A*). Animals in both treatment cohorts showed near-complete abrogation of tumor growth (p<0.01, one-way ANOVA, *Figure 5B and C*). Finally, the weight of residual tumors after 4 weeks of ME3BP-7 was significantly decreased in treated animals relative to controls (p=0.01, Mann-Whitney U test, *Figure 5D*).

We next tested the efficacy of ME3BP-7 in an orthotopic xenograft model generated with patient-derived PDAC TM01212 tumor pieces (*Figure 6A*). Orthotopic TM0212 xenografts expressed moderately high levels of both MCT1 RNA (37 transcripts per million) and protein (*Figure 6B*). Histological evaluation revealed the development of duct-like structures within these xenografts (*Figure 6B*). More importantly, all control animals developed metastases to the lung and liver, mimicking the behavior of human PDACs (*Figure 6F–H*). However, only 3/7 mice treated with ME3BP-7 had any distant metastases, and the number and size of metastases in the treated vs. untreated mice were significantly decreased in treated animals relative to controls (p<0.01, Mann-Whitney U test, *Figure 6H*). Again, histological evaluation of major organs did not reveal any pathological changes of significance as a result of potential toxicity in the treated mice (*Figure 6—figure supplement 1* and *Supplementary file 1*).

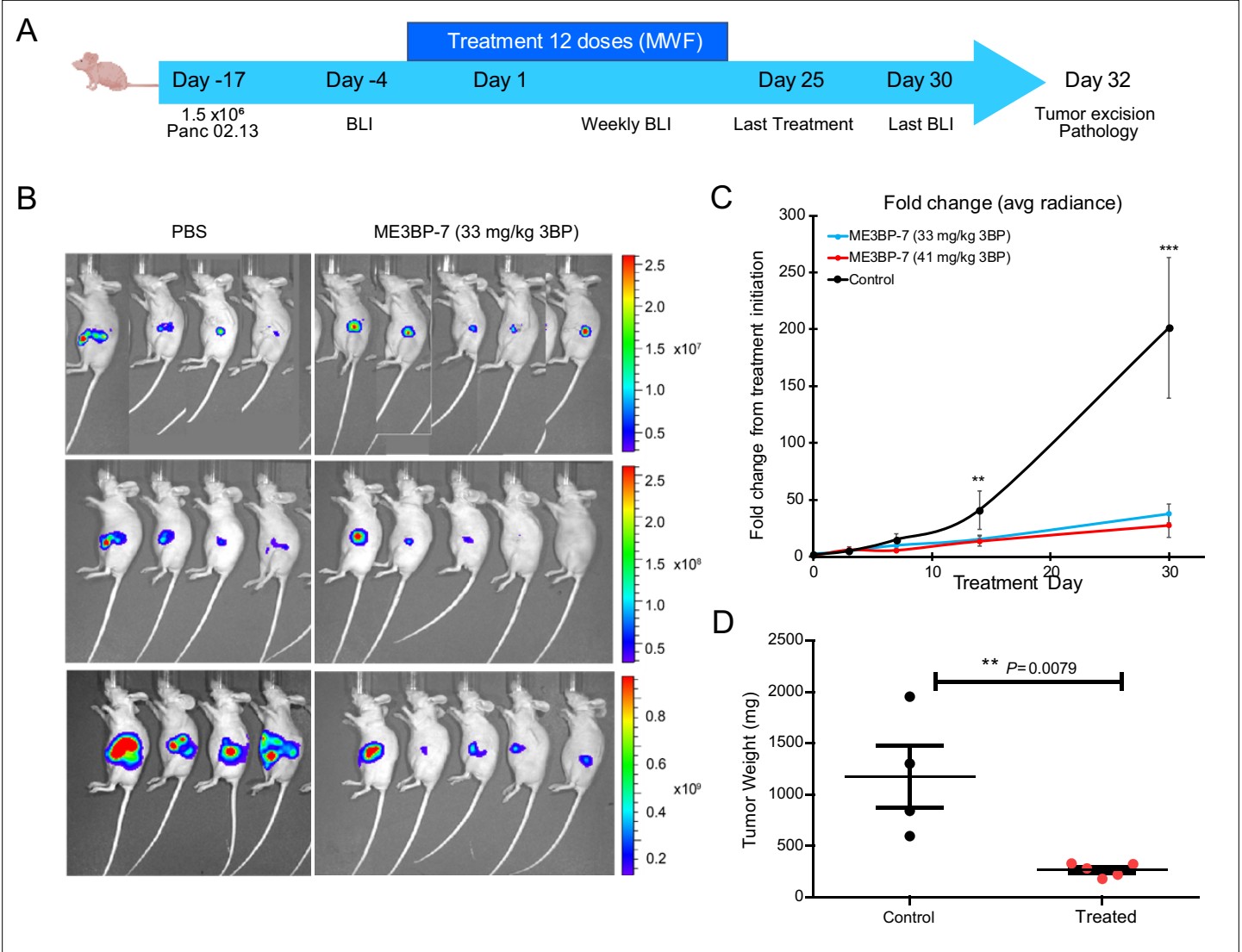

**Figure 5.** ME3BP-7 inhibits tumor growth of orthotopically implanted pancreatic cancer cell line Panc 02.13 with high monocarboxylate transporter 1 (MCT1) expression. (**A**) Timeline and design of in vivo tumor experiments. Created using BioRender.com. (**B**) Bioluminescence images of nude mice-bearing orthotopic Panc 02.13 tumors. (**C**) Mean fold change in radiance from day of treatment initiation (**p<0.01, ***p<0.001, one-way ANOVA). (**D**) Weights of residual tumors harvested upon termination of therapy (**p<0.01, Mann-Whitney U test). BLI (BioLuminescence Imaging).

The online version of this article includes the following figure supplement(s) for figure 5:

**Figure supplement 1.** Body weight changes over the course of ME3BP-7 administration in various murine models (**A**) Panc 02.13 in nude mice, (**B**) TM01212 in NSG mice, and (**C**) TM01098 in NCG mice.

We evaluated the efficacy of ME3BP-7 in a second PDAC patient-derived xenograft model, TM01098 (*Figure 6—figure supplement 2A*). Although these tumor cells expressed high levels of *MCT1* RNA (96 transcripts per million), protein expression exhibited focal staining (*Figure 6—figure supplement 2B*), which was in contrast to the uniform expression in Panc 02.13 (*Figure 1C*) and TM01212 xenografts (*Figure 6B*). Despite focal MCT1 protein expression, significantly slower growth of TM01098 xenografts occurred in animals treated with ME3BP-7 compared to controls, based on ultrasound (US) (p<0.01, Mann-Whitney U test, *Figure 6—figure supplement 2D*) and tumor weight (p=0.0028, *Figure 6—figure supplement 2E*).

## Analysis of expression of MCT1 in human tissues

To further validate the mRNA expression data in human tumors in publicly available databases (*Cancer Genome Atlas Research Network, 2017*; *Chen et al., 2019b*, *Figure 7—figure supplement 1*),

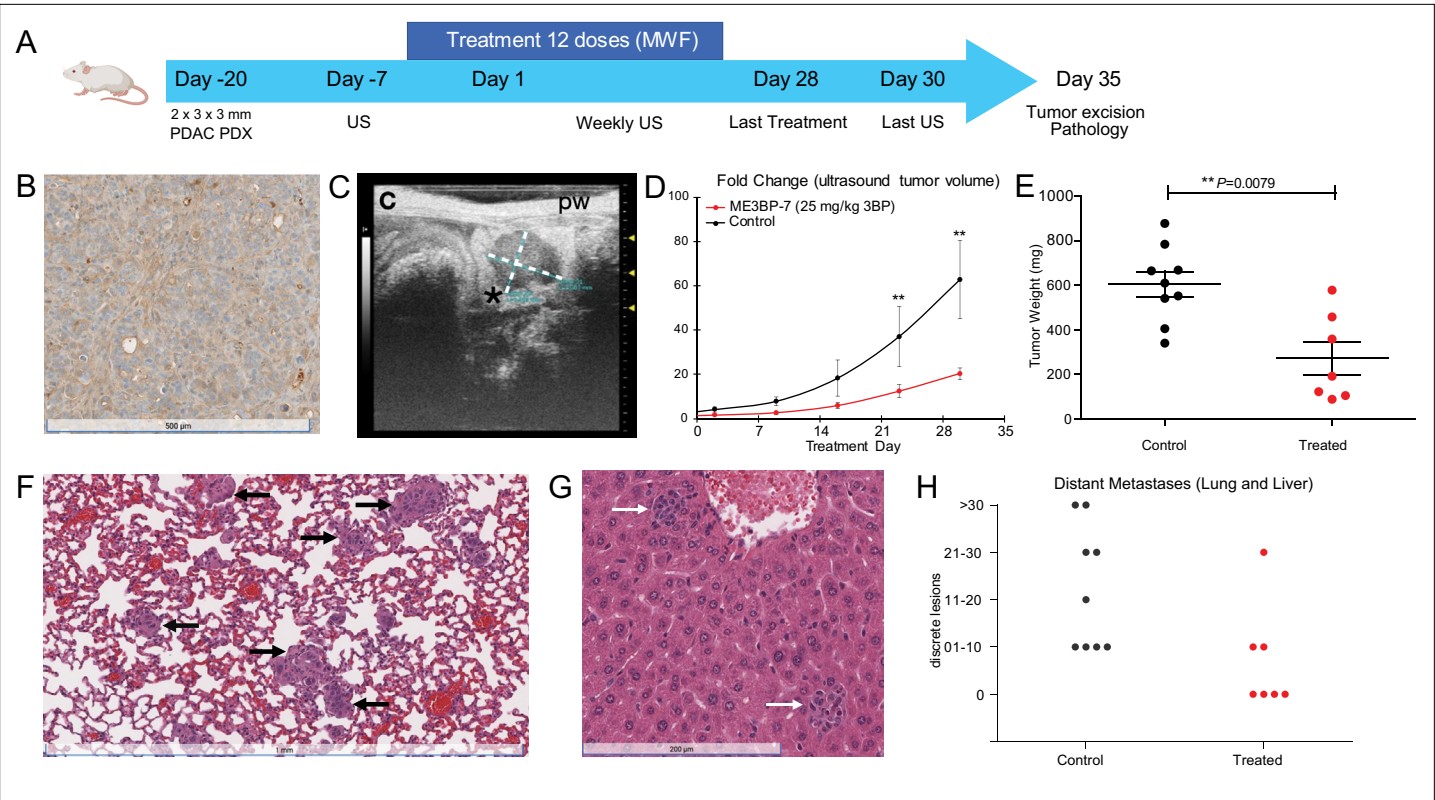

**Figure 6.** ME3BP-7 reduces tumor burden in orthotopically implanted human patient-derived xenograft TM01212 with diffuse expression of monocarboxylate transporter 1 (MCT1). (**A**) Timeline and design of in vivo therapeutic study. Created using BioRender.com. (**B**) Immunohistochemical analyses of orthotopic PDX TM01212 showing diffuse but uniform expression of MCT1. (**C**) Representative ultrasound image of orthotopically implanted tumors in NSG mice. (**D**) Mean fold change in tumor volume (n=10) from day of treatment initiation. (**E**) Weights of residual tumors harvested upon termination of therapy. (**F, G**) H&E of lung and liver with metastases from untreated animals. (**H**) Number of metastatic lesions harvested from control and treated mice upon termination of therapy (**\*\***p<0.01, Mann-Whitney U test). PDX (patient derived xenograft), US (ultrasound) MWF (Monday, Wednesday, Friday).

The online version of this article includes the following figure supplement(s) for figure 6:

**Figure supplement 1.** Representative H&E images of organs of NSG mice treated with ME3BP-7 for 4 weeks: (**A**) heart, (**B**) lung, (**C**) kidney, (**D**) pancreas, (**E**) liver, and (**F**) spleen ×10.

**Figure supplement 2.** ME3BP-7 reduces tumor burden in orthotopically implanted human patient-derived xenografts from a pancreatic ductal adenocarcinoma (PDAC) metastatic site with focally expressed monocarboxylate transporter 1 (MCT1) (TM01098).

immunohistochemistry was performed on a total of 95 PDACs on tissue arrays. Of the 95 samples analyzed, 56% (n = 53) were positive for MCT1 and 44% (n = 42) were negative. The staining intensity of the 53 MCT1 positive pancreatic cancer tissues was scored as follows: 58% (n = 31), low intensity; 30% (n = 16), intermediate intensity; and 12% (n = 6) high intensity (*Figure 7A and C*).

None of the 51 normal tissues recorded in the GTEx database (https://gtexportal.org/home/gene/SLC16A1) expressed average levels of *SLC16A1* RNA greater than the approximate threshold required for sensitivity to 3BP in cancer cell lines (~50 transcripts per million, *Figure 7—figure supplement 2*). The highest expression levels in normal tissues were found in the colon, testis, and uterus (median 37, 38, and 32 transcripts per million, respectively, among 142–376 tissue samples assessed for each tissue type) (*Uhlén et al., 2015*). The RNA levels were consistent with the immunohistochemistry; staining was low or absent in normal pancreas, brain, or ovary while readily observed in testis, colon, and uterine tissue (*Figure 7—figure supplement 3*). On the basis of this information, we carefully examined ME3BP-7-treated mice at necropsy and found no discernible damage after 4 weeks of treatment in either the colon or uterus, while the pancreatic cancers showed extensive necrosis (*Supplementary file 1*).

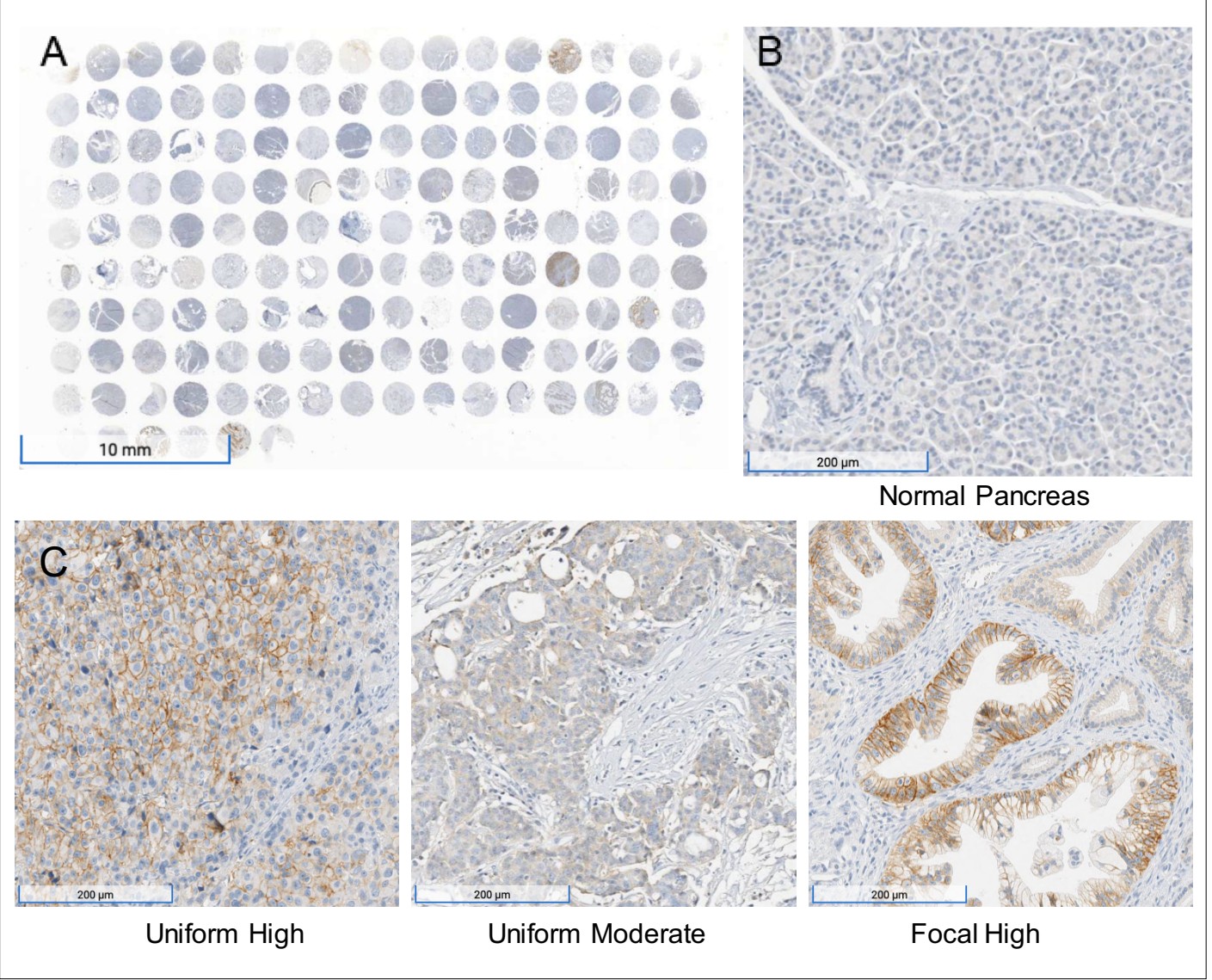

**Figure 7.** Immunohistochemistry performed on a pancreatic ductal adenocarcinoma (PDAC) tissue microarray with monocarboxylate transporter 1 (MCT1) antibody. (**A**) Overview (1×) of immunohistochemical (IHC) analyses performed on tissue microarray (HPanA150CS03, BioMax US) of pancreatic carcinoma cases with MCT1 antibody. (**B**) IHC of normal pancreas (10×) from the same microarray with MCT1 antibody. (**C**) Representative examples of uniform high, uniform moderate, and focal high expression of MCT1 in human PDAC samples from the array (10×).

The online version of this article includes the following figure supplement(s) for figure 7:

**Figure supplement 1.** Violin plot of TCGA data for pancreatic ductal adenocarcinomas (PDACs).

**Figure supplement 2.** Genotype-Tissue Expression (GTEx) dataset showing expression of monocarboxylate transporter 1 (MCT1) across 51 tissue types.

**Figure supplement 3.** Immunohistochemical (IHC) analyses of normal tissues from human TMA BC001130 (10×).

## Discussion

Pancreatic cancer is a major cause of cancer morbidity and mortality and is predicted to be the third leading cancer killer in the next decade. Average life expectancy is under a year in advanced disease, and there is an undeniable unmet need for better therapeutics. Public datasets reveal that a sizable fraction of pancreatic cancers express relatively high levels of a membrane protein, MCT1, (*Cancer Genome Atlas Research Network, 2017*), while the same is barely detectable in normal pancreas. Despite the promise, specific small-molecule inhibitors of MCT1 have not shown any benefit in human clinical trials, rendering MCT1 an unrealized but exciting biological target.

The results described in this study offer an innovative therapeutic approach using ME3BP-7 to target MCT1. The closest successful precedents for ME3BP-7 are antibody-drug conjugates such as trastuzumab emtansine (*Barok et al., 2014*) or inotuzumab ozogamicin (*Shor et al., 2015*). ME3BP-7 differs from antibody-drug conjugates in several important respects. First, ME3BP-7 is a small molecule, drastically reducing drug production challenges and cost. Second, ME3BP-7 itself mediates the targeting as well as the toxicity, while antibody-drug conjugates use an antibody for targeting and a separate, small molecule to kill cells (*Thomas et al., 2016*; *Baah et al., 2021*; *Drago et al., 2021*).

One of the most challenging aspects of the current study was the delivery of ME3BP-7 to the mice. After multiple attempts to deliver various 3BP formulations through other routes, we were only able to reliably deliver multiple doses of the drug intravenously (i.v.), and the number of injections and time periods over which we could administer the drug were limited. Importantly, this logistical issue is peculiar to small animals such as mice, in which repeated vascular access is difficult. In humans, delivery of anticancer drugs through pump-driven i.v. or intra-arterial catheters is routinely practiced (*Taxbro and Chopra, 2021*; *Schiffer et al., 2013*; *Sousa et al., 2015*).

Although we focused on pancreatic cancers in this study, MCT1 is overexpressed in subsets of mesotheliomas, leukemias, and several other tumor types (https://www.proteinatlas.org/ ENSG00000155380-SLC16A1/pathology). As with any targeted agent, we expect one important mechanism of resistance will be loss of the target (*Kobayashi et al., 2005*; *Shah and Fry, 2019*; *Garrett and Arteaga, 2011*). With targets that are the products of oncogenes, simple loss will not suffice, as the oncogene protein product is necessary for cell proliferation. In such cases, mutations in other sites of the protein that make it resistant to the drug, or other genes in the same pathway, generally occur (*Yu et al., 2013*; *Bean et al., 2008*; *Gambacorti-Passerini et al., 2003*; *von Bubnoff et al., 2002*). MCT1 is not an oncogene, so loss of MCT1 is conceivable.

The results in *Figure 3* show that ME3BP-7 is uniquely fast-acting compared to other drugs now used in the clinic to treat patients with pancreatic cancer. Even brief exposure to ME3BP-7 results in effective cell killing of the majority of cells. On the other hand, a limitation of our study is that, despite these striking effects in vitro, we were not able to induce true regressions of extant tumors in mice, though growth was slowed (*Figures 5 and 6*, *Figure 6—figure supplement 2*), and metastases were markedly reduced (*Figure 6H*). We speculate that different methods for administering ME3BP-7 may improve its in vivo potency. In addition, potentially synergistic combinations with new and existing agents could also be explored to augment the potency as well as broaden the therapeutic potential of ME3BP-7. Our data suggests that the combination of irinotecan, or liposomal irinotecan as described in the NAPOLI-3 trial (*Wang-Gillam et al., 2019*; *Wainberg et al., 2023*), with ME3BP-7 could improve the potency as well as address the pockets of heterogeneity with low MCT1 in pancreatic cancer (*Figure 4*, *Figure 4—figure supplement 1*). Similarly, ME3BP-7 could alleviate the resistance to glutaminase inhibition, leading to an efficacious combination (*Ammar et al., 2023*). Finally, severe extracellular acidity around solid tumors, due to MCT1 overexpression causing immune exclusion (*Feng et al., 2024*), suggests a combination of immune checkpoint inhibitors with ME3BP-7 could improve therapeutic outcomes. However, given the limitations of administering the drug to small animals, we believe that clinical trials will be the only way to reliably determine whether ME3BP-7 could serve as an adjuvant therapy for patients with pancreatic cancers. Our results highlight that ME3BP-7 shows efficacy across multiple pancreatic cancer model systems and potentially represents a powerful new therapeutic agent for this cancer type and perhaps others that express the drug's target protein.

## Materials and methods
### Study design
We designed a new formulation of the cytotoxic agent 3BP to be resistant to serum degradation and systemically administered. Modified cyclodextrins were investigated for their efficiency of encapsulating 3BP, and the activity of the encapsulated drug was analyzed after exposure to serum. We used CRISPR-mediated technologies to knock out the cell membrane receptor for 3BP, MCT1, in pancreatic cancer cell lines in vitro. We also modified pancreatic cancer cell lines to express fluorescent or bioluminescent markers for the purposes of tracking cellular responses to ME3BP-7 as well as other drugs commonly used in the clinic for treating PDAC patients. Tumor cells or tumor fragments were orthotopically implanted into the pancreas of mice for in vivo experiments. Implantation was

confirmed with luciferase expression in cell lines or US, followed by stratification and randomization into treatment and control groups. The MCT1-dependent killing potential of ME3BP-7 was measured with time-lapse recordings obtained with an IncuCyte Live Cell-Analysis System (Essen Bioscience; Ann Arbor, MI, USA). Necropsies were performed for tumor resection and organ evaluation at the end of the experiment. Sample sizes for animal experiments were selected based on previous experience with the animal models but were not predetermined by power analysis. No animals were excluded from the study due to illness unless indicated. The number of replicates in each experiment is noted in the figure legends.

**Key resources table**

| Reagent type (species) or resource | Designation | Source or reference | Identifiers | Additional information |
|---|---|---|---|---|
| Strain, strain background (*Mus musculus*, female) | *M. musculus* NOD-Prkdc$^{em26Cd52}$Il2rg$^{em26Cd22}$/NjuCrl (NCG) | Charles River GmbH | | |
| Strain, strain background (*M. musculus*, female) | *M. musculus* Crl:NU(NCr)-Foxn1nu(Athymic nude) | Charles River GmbH | | |
| Strain, strain background (*M. musculus*, female) | *M. musculus* NOD.Cg-Prkdcscid Il2rgtm1Wjl/SzJ (NSG) | Jackson Laboratories | | |
| Cell line (*Homo sapiens*) | HEK293T | ATCC | Cat #CRL-3216 | |
| Cell line (*H. sapiens*) | MIA PaCa-2 | ATCC | Cat #CRL-1420 | |
| Cell line (*H. sapiens*) | Panc 02.13 | ATCC | Cat #CRL-2554 | |
| Cell line (*H. sapiens*) | PSN-1 | ATCC | Cat #CRL-3211 | |
| Cell line (*H. sapiens*) | AsPC-1 | ATCC | Cat # CRL-1682 | |
| Cell line (*H. sapiens*) | BxPC-3 | ATCC | Cat # CRL-1687 | |
| Cell line (*H. sapiens*) | CFPAC-1 | ATCC | Cat #CRL-1918 | |
| Cell line (*H. sapiens*) | DLD-1 | ATCC | Cat #CCL-21 | |
| Cell line (*H. sapiens*) | MIA PaCa-2 MCT1 KO | This study | N/A | |
| Cell line (*H. sapiens*) | DLD-1 MCT1 KO | This study | N/A | |
| Cell line (*H. sapiens*) | MIA Paca-2 red | This study | N/A | |
| Cell line (*H. sapiens*) | MIA PaCa-2 MCT1 KO green | This study | N/A | |
| Cell line (*H. sapiens*) | DLD-1 MCT1 KO | This study | N/A | |
| Cell line (*H. sapiens*) | Panc 02.13 luc | This study | N/A | |
| Cell line (*H. sapiens*) | PSN-1 luc | This Study | N/A | |
| Cell line (*H. sapiens*) | Pancreatic cancer patient derived xenograft | Jackson Laboratories | TM01098 | |
| Cell line (*H. sapiens*) | Pancreatic cancer patient derived xenograft | Jackson Laboratories | TM01212 | |
| Antibody | Mouse anti MCT1 monoclonal | Santa Cruz Biotech | sc-365501 | |
| Recombinant DNA reagent | Firefly Luciferase lentivirus | Cellomics Tech | PLV-10003–50 | |
| Recombinant DNA reagent | Incucyte Nuclight Green Lentivirus (puro) | Sartorius | 4624 | |
| Recombinant DNA reagent | Incucyte Nuclight Red Lentivirus (puro) | Sartorius | 4625 | |
| Sequence-based reagent | Alt-R CRISPR Cas9 crRNAs (ACCATGCCATTCAGGCTAGT) | IDT | N/A | |
| Sequence-based reagent | and Alt-R CRISPR-Cas9 tracrRNA | IDT | 1072532 | |
| Peptide, recombinant protein | Cas9 Nuclease | IDT | 10081059 | |
| Commercial assay or kit | Agilent DNA ScreenTape | Agilent | Cat #5067–5576 | |
| Commercial assay or kit | Agilent DNA ScreenTape Sample Buffer | Agilent | Cat #5067–5577 | |

*Continued on next page*

*Continued*

| Reagent type (species) or resource | Designation | Source or reference | Identifiers | Additional information |
|---|---|---|---|---|
| Commercial assay or kit | Agilent DNA Ladder | Agilent | Cat #5067–5578 | |
| Commercial assay or kit | Luciferase Assay System | Promega | Cat #E1501 | |
| Chemical compound, drug | 1% Penicillin-Streptomycin | Thermo Fisher Sci | Cat #15140122 | |
| Chemical compound, drug | 4–15% Mini-PROTEAN TGX Precast Protein Gels | Bio-Rad | Cat #456–1086 | |
| Chemical compound, drug | 5-Fluoruracil | Selleck chem | Cat #S1209 | |
| Chemical compound, drug | Gemcitabine | Selleck chem | Cat #S1149 | |
| Chemical compound, drug | Irinotecan hydrochloride | Selleck chem | Cat #S5026 | |
| Chemical compound, drug | Oxaliplatin | Selleck chem | Cat #S1224 | |
| Chemical compound, drug | Bromopyruvic acid | Sigma Aldrich | Cat #16490–10 G | |
| Chemical compound, drug | Succinyl-β-cyclodextrin | Sigma Aldrich | Cat #85990–5 G | |
| Chemical compound, drug | 2-Hydroxypropyl-β-cyclodextrin | Sigma Aldrich | Cat #778966–100 G | |
| Chemical compound, drug | Dimethyl Sulfoxide (DMSO) | Sigma Aldrich | Cat #C6295 | |
| Chemical compound, drug | Fetal Bovine Serum (FBS) | HyClone | Cat #16777–006 | |
| Chemical compound, drug | RPMI 1640 medium | Gibco | Cat #11875–119 | |
| Chemical compound, drug | DMEM medium | Gibco | Cat #11995065 | |
| Chemical compound, drug | EMEM medium | ATCC | Cat #30–2003 | |
| Chemical compound, drug | EPITHELIAL CELL MEDIUM-Complete Kit | Science Cell Research | Cat #4101 | |
| Chemical compound, drug | Phosphate Buffered Saline (PBS) | Thermo Fisher | Cat #J60465.K2 | |
| Chemical compound, drug | Glycerol | Sigma Aldrich | Cat #G5516 | |
| Chemical compound, drug | Phusion Flash High-Fidelity PCR Master Mix | Thermo Fisher | Cat #F548S | |
| Chemical compound, drug | Pierce ECL Western Blotting Substrate | Thermo Fisher | Cat #32106 | |
| Chemical compound, drug | Protease inhibitor | Millipore Sigma | Cat #4693159001 | |
| Chemical compound, drug | RediJect D-Luciferin Ultra Bioluminescent Substrate | PerkinElmer | Cat #770505 | |
| Chemical compound, drug | SsoAdvanced Universal SYBR Green Supermix | Bio-Rad | Cat #1725270 | |
| Chemical compound, drug | Trypsin | Gibco | Cat #25300054 | |
| Other | Tissue Microarray | US Biomax, Inc | BC001130 | |
| Other | Tissue Microarray | US Biomax, Inc. | HPanA150CS03 | |

Further information and requests for resources and reagents should be directed to and will be fulfilled by the lead contact, Surojit Sur (ssur1@jhmi.edu). Isogenic knockout and engineered cell lines are available through The Genetic Resources Core Facility at Johns Hopkins School of Medicine (jhbiobank@jhmi.edu, MD, USA).

## Synthesis of encapsulated 3BP

Drug batches were generated as previously described. In brief, a solid sample of 3BP (1 mmol, 167 mg) was added in small portions to a solution of the corresponding cyclodextrin (1.0–1.2 mmol) in 25 mL of deionized (DI) water under constant agitation. After complete addition over 15–20 min, the samples were further agitated for 1–4 hr at 25°C. Finally, the samples were flash-frozen in liquid nitrogen or dry ice/acetone and lyophilized overnight. The lyophilized reagent was diluted in PBS 15 and 30 min prior to in vitro and in vivo experiments, respectively.

## Size-exclusion HPLC chromatography

SEC was performed with a Shodex-OH Pak at an elution rate of 1 mL/min of PBS under isocratic conditions and monitored at 220 nm. The samples included (i) free 3BP (1 mg/mL); (ii) succinyl-β-CD without 3BP (20 mg/mL); (iii) a mixture of 10 μL of free 3BP at 1 mg/mL and 10 μL ME3BP-7 at 10 mg/mL; and (iv) ME3BP-7 (10 mg/mL).

## Cell culture

Human PDAC cell lines, including MIA PaCa-2, PSN-1, Panc 02.13, AsPC-1, BxPC-3, and CFPAC-1, were obtained from the American Type Cell Culture (ATCC) (Manassas, VA, USA), authenticated by STR and tested for mycoplasma. Cell lines were maintained at 37°C in a humidified 5% $CO_2$ atmosphere in T75 tissue culture flasks containing 14 mL of the medium recommended by ATCC (DMEM for MIA PaCa-1 parental and MCT1 KO; RPMI 1640 for Panc 02.13, AsPC-1, BxPC-3, and DLD-1; and IMDM for CFPAC-1). All media were supplemented with 10% fetal bovine serum (Mediatech, Inc; Manassas, VA, USA) and 1 % penicillin-streptomycin. Cells were passaged at the ATCC recommended ratio every 4–5 days with trypsinization and resuspended in fresh medium in a new flask. Only early passaged cells harvested at 70–85% confluence were used for in vitro and in vivo experiments.

## Lentiviral transduction

MIA PaCa-2 or Panc 02.13 cells were transduced with a CMV-Firefly luciferase lentivirus carrying a puromycin-selectable marker (Cellomics Tech; Halethorpe, MD, USA) according to the manufacturer's instructions. Parental and MCT1 KO MIA PaCa-2 cells were transduced with Lentiviral NucLight Red and Green vectors (Sartorius), respectively, carrying a puromycin-selectable marker. All transduced cells were selected in growth medium containing puromycin (4 μg/mL) for 96 hr.

## Viability assays

Cells (3–12 × 10³) were seeded onto the wells of 96-well plates and exposed to drug 24 hr after plating. For MIA PaCa-2 parental and MCT1 KO co-culture experiments, cells were suspended in medium at equal concentration and seeded at 12,000/genotype onto wells. Cells in experiments were imaged with time-lapse fluorography using the IncuCyte Live Cell-Analysis System (Essen Bioscience; Ann Arbor, MI, USA). For green- and red-labeled cells, a processing definition for the IncuCyte was created, and the green or red fluorescence channels were used for analysis. All in vitro experiments were conducted at least in duplicate. Green and red object counts were used to track parental and MCT1 KO MIA PaCa-2 experiments, while confluence was used to track unlabeled cells. Cell viabilities were recorded as the fraction of cells that survived treatment at the indicated drug molarities relative to the control wells. Dose-response curves were fit using Prism 9 software.

## Serum stability assays

Free 3BP, ME3BP-7, and HPCD-3BP were incubated at various concentrations (12.5 μM, 25 μM, 50 μM, 100 μM, and 200 μM) with 90 μL of human serum (Sigma, Cat No. H3667; St. Louis, MO, USA) at 37°C for up to 8 hr. Aliquots were collected at 30 min, 1 hr, 2 hr, 4 hr, and 8 hr and stored at –80°C until further analysis. For the cell toxicity assay used to assess the amount of biologically active 3BP at each time point, parental DLD-1 (MCT1, 387 TPM) cells were plated at 35–40% confluence and exposed to 10 μL of the collected drug sample diluted with 190 μL of medium. Cell death was measured by imaging cells after 72 hr in culture with the IncuCyte Live Cell-Analysis System (Essen Bioscience). Residual drug activity was estimated by comparison of cell death induced by drug samples incubated with human serum for different periods of time to the cell death induced by an unincubated drug sample to derive the % drug activity.

## Timed exposures to the drug

Cells were seeded onto 96-well plates at a density of 24,000 cells per well (12,000 parental MIA PaCa-2 red and 12,000 MCT1 KO MIA PaCa-2 green). After 24 hr, cells were treated with vehicle (complete DMEM) or a serial dilution of a drug (bromopyruvic acid, Sigma-Aldrich; gemcitabine HCl, irinotecan, oxaliplatin, and 5-fluorouracil, Selleck Chemicals; Houston, TX, USA). Each treatment condition was conducted in triplicate unless otherwise stated. The drugs were either removed from the plates at 30 min or 2 hr, or left on cells. Wells were gently washed with 200 μL of PBS twice and replaced with fresh

medium. Residual cells were assessed with IncuCyte Live Cell-Analysis System. Cell viabilities were reported as the percentage of cells remaining after treatment at the indicated drug molarities relative to the average of three control wells without drug.

## Genetic inactivation of *SLC16A1* (MCT1)

The Alt-R CRISPR system (Integrated DNA Technologies [IDT]; Coralville, IA, USA) was used to delete *SLC16A1*, the gene encoding MCT1 protein, in the DLD-1 and MIA PaCa-2 cell lines. The gRNA sequence was designed with CHOPCHOP v3 (*Labun et al., 2019*). Alt-R CRISPR Cas9 crRNAs (ACCA TGCCATTCAGGCTAGT, IDT; SEQ ID NO:1) and Alt-R CRISPR-Cas9 tracrRNA (1072532, IDT) were resuspended in Nuclease-Free Duplex Buffer (IDT) at a concentration of 100 µM. The crRNAs and tracrRNA were mixed in a 1:1 molar ratio and denatured for 5 min at 95°C, followed by slow cooling to room temperature for duplexing. Cas9 Nuclease (1081059, IDT) was then added at a 1.2:1 molar ratio and allowed to stand at room temperature for 15 min. Forty pmoles of the Cas9 ribonucleoprotein containing tracrRNA/MCT1 crRNA duplex was mixed with $2 \times 10^5$ cells in 20 µL of OptiMEM (31985088, Thermo Fisher Scientific; Waltham, MA, USA), loaded into a 0.1 cm cuvette (1652089, Bio-Rad; Hercules, CA, USA), and electroporated at 120 V for 16 ms using an ECM 2001 (Harvard Apparatus; Holliston, MA, USA). Cells were immediately transferred to complete growth medium and cultured for 1 week. Upon reaching confluence, cells were plated at a density of 0.5–2 cells per well in 96-well plates and cultured for 3 weeks. Single colonies were transferred into 2 replica 96-well plates. Genomic DNA was harvested from one of the plates using the Quick-DNA 96 Kit (Zymo Research; Orange, CA, USA) and PCR-amplified using Q5 Hot Start High-Fidelity 2X Master Mix (New England BioLabs; Ipswich, MA, USA). SafeSeqS (*Kinde et al., 2011*) was used to confirm the mutation status of selected clones. Single MIA PaCa-2 clones deleted for *SLC16A* were collected from the matched replica plate, and six clones were pooled at equal ratios for further experiments.

## MCT1 immunostaining

DLD-1 parental and isogenic DLD-1-MCT1 KO cells were used as the positive and negative controls for testing the specificity of five commercially available MCT1 antibodies. Cells were trypsinized, washed in media to inactivate the trypsin, pelleted at 400×*g* for 10 min, fixed in 10% formalin, and embedded in paraffin. Sections (4 µm) were cut from the blocks and incubated with different dilutions of primary and species-appropriate secondary antibodies. The anti-MCT1 mouse antibody from Santa Cruz Biotechnology (SC-365501, Lot number D2319; Dallas, TX, USA) was chosen for immunostaining with a Ventana Discovery Ultra autostainer (Roche Diagnostics; Indianapolis, IN, USA) at the Oncology Tissue Services Core of Johns Hopkins University School of Medicine. Briefly, following deparaffinization and rehydration of sections, epitope retrieval was performed in Ventana Ultra CC1 buffer (catalog# 6414575001, Roche Diagnostics) at 96°C for 64 min. Antibodies were diluted in antibody dilution buffer (catalog# 5280524001, Roche Diagnostics; anti-MCT1 antibody, 1:2000 for cell pellets, 1:200 for patient-derived xenografts, and 1:100 for other tissues or tissue microarrays) and incubated with the slides at 36°C for 60 min. Following standard washing in the Ventana auto-stainer, bound antibodies were detected with an anti-mouse HQ detection system (catalog# 7017936001 with 7017782001, Roche Diagnostics) and a Chromomap DAB IHC detection kit (catalog # 5266645001, Roche Diagnostics). The slides were then counterstained with Mayer's hematoxylin, dehydrated, and mounted in Toluene mounting medium (MER 7720, Mercedes Scientific; Lakewood Ranch, FL, USA).

## Tissue microarrays

IHC for MCT1 was performed on two commercially available FFPE tissue microarrays (BC001130 and HPanA150CS03; US Biomax, Inc; Rockville, MD, USA) for a total of 100 cases of pancreatic cancer. BC001130 included 20 cases of pancreatic carcinoma (triplicate cores per case), and HPanA150CS03 included 80 cases with adjacent normal tissue. Two independent reviewers, including a pathologist, scored the intensity and pattern of MCT1 staining for tumor and normal tissues. Any differences in grading were reviewed and resolved. Upon evaluation, 5 samples were considered to be of poor quality and disregarded. Of the 95 samples analyzed, 56% (n = 53) were positive for MCT1 and 44% (n = 42) were negative. The staining intensity of the 53 MCT1 positive pancreatic cancer tissues was scored as follows: 58% (n = 31), low intensity; 30% (n = 16), intermediate intensity; and 12% (n = 6) high intensity.

## Animal experiments

Escalating doses ranging from 8.0 to 33 mg/kg of free 3BP (as ME3BP-7) were infused via the tail vein of healthy nude mice to determine the maximum tolerated dose. Repeated tail vein injections of free 3BP caused significant scarring of the tail, limiting the use for sustained injections of the free drug, while injections of ME3BP-7 exhibited minimal sclerosis. For optimal comparison with the free drug, implantable VABs were used to provide direct access to the jugular vein for accurate administration of both drugs. Crl:NU(NCr)-$Foxn1^{nu}$ (athymic nude mice; Charles River (490); Wilmington, MA, USA; *Figure 5*) were used for experiments performed with pancreatic tumor cell lines. More severely, immunodeficient mice were used for patient-derived xenografts which did not uniformly grow in athymic nude mice. In addition, different types of severely immunocompromised mice were used for the patient-derived xenografts due to limited availability of animals during the COVID pandemic: NOD.Cg-Prkdc Il2rg /SzJ ((NSG), the Jackson Laboratory (005557); Bar Harbor, ME, USA; *Figure 6*); and NOD-$Prkdc^{em26Cd52}Il2rg^{em26Cd22}$/NjuCrl (NCG) (Charles River (572)); *Figure 5—figure supplement 1*.

## Models

### Panc02.13

Orthotopic tumors were generated by implanting $1.5 \times 10^6$ luciferase-expressing Panc 02.13 cells (with 10% Matrigel) in the pancreas of 20 nude mice. On day 13 postimplantation, tumor burden was assessed with the IVIS live-cell imaging system, and 15 animals with similar bioluminescence signals were stratified and randomized into three cohorts. On day 16, a second baseline image was recorded, and treatment was initiated on day 17 postimplantation. Animals in the control group (n = 4) were administered 200 µL of PBS (vehicle), while the two treatment arms received 33 (n = 5) and 41 mg/kg (n = 4) of 3BP in the form of ME3BP-7 in 200 µL of PBS. Body weights were measured immediately before drug administration to ensure appropriate dosage and delivery. Treatments were administered by a single i.v. bolus delivered over 30 s for 4 weeks on each Monday, Wednesday, and Friday (total of 12 injections), and bioluminescence was recorded once each week. At the end of 4 weeks, all animals were euthanized, and their tumors weighed.

### TM01212

Subcutaneous TM01212 xenografts were harvested from 3 hosts, and tumor pieces (2 mm × 1 mm × 1 mm) were orthotopically transferred into the pancreas of 20 NSG mice. On day 13, tumor-bearing animals (take rate 90%, 18 mice) were identified with US and randomized into 2 groups- animals showing no tumors were excluded from the study. The control group was infused with the inactive agent (270 mg/kg sCD), while the treatment group was administered ME3BP-7 formulated at 25 mg/kg of 3BP. Bolus injections via the tail vein were administered every Monday, Wednesday, and Friday for 4 weeks (12 total doses) starting at day 20 post-tumor implantation. Tumor growth was tracked with weekly US measurements, and animals were euthanized on day 35 after the initiation of treatment to assess tumor burden in the pancreas, lung, and liver. Two of the treated mice received <80% of the intended dose of ME3BP-7 (due to tail vein sclerosis) and were removed from the analysis. Some mice did not receive the full dose due to tail scarring and responded less well than the other mice.

### TM01098

Subcutaneous TM01098 xenografts were harvested from 3 hosts, and tumor pieces (2 mm × 3 mm × 3 mm) were transplanted into 29 NCG mice (17 with surgically implanted VABs and 12 without). On day 9, tumor-bearing animals were identified with US (take rate 76%). Five mice died from VAB-related complications and were removed from the study. Tumor-bearing hosts were separated into two groups: the mice in the control group (n = 7) received 200 µL of PBS via tail vein i.v. bolus infusion (Monday, Wednesday, and Friday), while the mice in the treatment group (n=11) received 21 mg/kg 3BP as ME3BP-7 via a VAB on the same schedule. US measurements were taken weekly. Treatment started on postimplantation day 14, and the hosts were subjected to this regimen for 5 weeks. Mice underwent a final US on treatment day 38 and were euthanized. Tumor burden in the pancreas and at distal sites was assessed by necropsy.

## Surgeries

For orthotopic models, cells or small pieces of xenografts were implanted into the pancreas of each mouse in protocols modified from previously described techniques (*Erstad et al., 2018*). Once under anesthesia, the skin over the left abdomen was shaved and sterilized. A horizontal 1 cm incision was made over the left upper quadrant of the abdomen, right below the ribs. The underlying peritoneum was incised, and the spleen and pancreas were located. The pancreatic tissue was gently extruded from the abdominal cavity for good visualization. Either a cell aliquot was injected with a Hamilton syringe, or a piece of tumor was sutured to the tail of the pancreas with a 7-0 polypropylene suture avoiding major vessels. The implantation of the tumor fragment was standardized by size, time from extraction to implantation, and implantation technique in the tail of the pancreas. After implantation, the pancreas was gently returned to the abdominal cavity. The abdominal wall was closed with absorbable sutures, and the skin incision was closed using wound clips. Wound clips were removed once the incision site had healed, usually ~10 days after surgery.

Mice were euthanized upon termination of the experiment or when major weight loss or toxicity was observed according to JHU Animal Care and Use Committee standards. Tumors and organs were harvested, weighed, placed in 10% formalin, processed, fixed in paraffin, and sectioned. Standard H&E staining was performed on sections, and an expert comparative pathologist reviewed all presented data.

## In vivo imaging for orthotopic Panc 02.13 animal model

Luminescence quantification was performed using the IVIS imaging system and Living Image software (PerkinElmer; Waltham, MA, USA). Mice were anesthetized at 37°C with inhaled isoflurane in an induction chamber for 5 min and received an intraperitoneal injection of luciferin (150 mL, RediJect D-Luciferin Ultra Bioluminescent Substrate, PerkinElmer, 770505). Bioluminescence images were taken 13 min after injection, and control fluorescence images were acquired at each screening to confirm satisfactory intraperitoneal luminescence.

## US protocol for orthotopic PDX animal model

A modification of an US imaging method for pancreas in mice was used (*Ganapathy-Kanniappan et al., 2009*). The left flanks of mice were shaved with a clipper. Mice were then injected with 2 mL of 0.9% sterile saline intraperitoneally to increase the contrast between intrabdominal organs, anesthetized with isoflurane in the induction chamber for 5 min, and placed in the lateral recumbent position with the left flank up on the imager over a heated pad. Continuous anesthesia was applied through a face cone. Pancreatic tumors were detected and measured through acquisition of a high-resolution US obtained with the VisualSonics Vevo2100 High-Resolution Ultrasound System. A 15 mm depth US window in B-mode acquisition was used on all mice. After applying gel over the area for US visualization, images were obtained in a standardized fashion. The spleen, liver, left kidney, and pancreas were identified first to confirm correct positioning. Tumors were hypoechoic (dark/gray), while the surrounding pancreas was hyperechoic (bright/white). The tumor sutures were the most hyperechoic and frequently marked the tumor center. Trans-axial US images of the tumor were obtained by placing the US probe parallel and distal to the rib cage with the notched side of the transducer to the left (pointing anteriorly). Multiple anterior and posterior images were taken in this plane spanning the pancreas. Longitudinal US images were then obtained by placing the US probe parallel to the mouse axis with the notch side pointing toward the head of the mouse. Videos were acquired across the tumor and at the point where the maximum tumor diameter was visualized, and multiple images were saved for measurement for the longitudinal and the trans-axial views. For mice with several discrete tumors (e.g. at the peritoneal wall), the same process was repeated for each tumor. After imaging, mice were cleaned and allowed to recover completely from anesthesia before returning to their cage. To obtain measurements, files were loaded with the US software study management function. Maximal longitudinal and transversal diameters were obtained for each tumor image. To calculate tumor volumes, the formula $V=a*b^2/2$, where a is the largest and b is the smallest diameter within the 4 obtained measurements (2 trans-axial and 2 longitudinal) (*Tang et al., 2012*).

## Statistical methods

Data are presented as the mean ± SD unless otherwise specified. Statistical analyses were performed with the tests as indicated. A p-value of <0.05 was considered statistically significant unless otherwise indicated. All analyses and graph production were performed using Prism version 9.0 (GraphPad) or Microsoft Excel.

## Acknowledgements

We would like to thank Suman Paul, Sarah DiNapoli, and Tushar Nichakawade for discussions and thoughtful comments. The Virginia and DK Ludwig Fund for Cancer Research. Lustgarten Foundation for Pancreatic Cancer Research. The Commonwealth Fund. The Bloomberg~Kimmel Institute for Cancer Immunotherapy. JRT was supported by NIH Grant R25 5R25NS065729. BJM was supported by NCI grant T32 GM136577. CB was supported by NCI Grant R37 CA230400 and the Burroughs Wellcome Career Award for Medical Scientists.

## Additional information

### Competing interests

Chetan Bettegowda: is a consultant to Depuy-Synthes, Bionaut Labs, Haystack Oncology and Galectin Therapeutics; is also a co-founder of OrisDx and Belay Diagnostics. Nickolas Papadopoulos: is a founder of Thrive Earlier Detection, an Exact Sciences Company; is a consultant to Thrive Earlier Detection; holds equity in Exact Sciences; is a founder of, or consultants to, and own equity in ManaT Bio., Haystack Oncology, Neophore, Vidium, CAGE Pharma, and Personal Genome Diagnostics. Kenneth W Kinzler: is a founder of Thrive Earlier Detection, an Exact Sciences Company; is a consultant to Thrive Earlier Detection; holds equity in Exact Sciences; is a founder of, or consultants to, and own equity in ManaT Bio., Neophore, and Personal Genome Diagnostics; holds equity in Haystack Oncology and CAGE Pharma. Bert Vogelstein: is a founder of, or consultant of, or holds equity in Thrive Earlier Detection, Exact Biosciences, Clasp Therapeutics, Haystack Oncology, Neophore; is consultant and owns equity in Catalio Capital Management. Surojit Sur: is a consultant for CAGE Pharma. The other authors declare that no competing interests exist.

### Funding

| Funder | Grant reference number | Author |
| --- | --- | --- |
| Ludwig Family Foundation | | Jordina Rincon-Torroella<br>Marco Dal Molin<br>Brian Mog<br>Gyuri Han<br>Evangeline Watson<br>Nicolas Wyhs<br>Nickolas Papadopoulos<br>Kenneth W Kinzler<br>Shibin Zhou<br>Bert Vogelstein<br>Surojit Sur |
| Lustgarten Foundation | | Jordina Rincon-Torroella<br>Marco Dal Molin<br>Brian Mog<br>Gyuri Han<br>Evangeline Watson<br>Nicolas Wyhs<br>Shun Ishiyama<br>Taha Ahmedna<br>Nickolas Papadopoulos<br>Kenneth W Kinzler<br>Shibin Zhou<br>Bert Vogelstein<br>Kathleen Gabrielson<br>Surojit Sur |

| Funder | Grant reference number | Author |
|---|---|---|
| Commonwealth Fund | | Jordina Rincon-Torroella<br>Marco Dal Molin<br>Brian Mog<br>Gyuri Han<br>Evangeline Watson<br>Nicolas Wyhs<br>Shun Ishiyama<br>Taha Ahmedna<br>Il Minn<br>Nilofer Azad<br>Chetan Bettegowda<br>Nickolas Papadopoulos<br>Kenneth W Kinzler<br>Shibin Zhou<br>Bert Vogelstein<br>Kathleen Gabrielson<br>Surojit Sur |
| National Institutes of Health | R25: 5R25NS065729 | Jordina Rincon-Torroella |
| National Cancer Institute | T32: GM136577 | Brian Mog |
| National Cancer Institute | R37: CA230400 | Chetan Bettegowda |
| Burroughs Wellcome Fund | Career Award | Chetan Bettegowda |
| Bloomberg~Kimmel Institute for Cancer Immunotherapy | | Chetan Bettegowda<br>Nickolas Papadopoulos<br>Kenneth W Kinzler<br>Shibin Zhou<br>Bert Vogelstein<br>Surojit Sur |

The funders had no role in study design, data collection and interpretation, or the decision to submit the work for publication.

## Author contributions

Jordina Rincon-Torroella, Validation, Investigation, Visualization, Methodology, Writing – original draft, Project administration, Writing – review and editing; Marco Dal Molin, Validation, Investigation, Visualization, Methodology, Project administration, Writing – review and editing; Brian Mog, Evangeline Watson, Validation, Investigation, Methodology; Gyuri Han, Investigation, Visualization, Methodology; Nicolas Wyhs, Taha Ahmedna, Investigation, Methodology; Shun Ishiyama, Software, Visualization; Il Minn, Resources, Validation, Visualization; Nilofer Azad, Validation, Investigation, Visualization, Methodology, Writing – review and editing; Chetan Bettegowda, Supervision, Validation, Visualization; Nickolas Papadopoulos, Shibin Zhou, Supervision, Validation, Visualization, Writing – original draft, Writing – review and editing; Kenneth W Kinzler, Supervision, Validation, Visualization, Methodology, Writing – original draft, Project administration, Writing – review and editing; Bert Vogelstein, Supervision, Funding acquisition, Validation, Investigation, Visualization, Methodology, Writing – original draft, Writing – review and editing; Kathleen Gabrielson, Supervision, Validation, Investigation, Visualization, Methodology, Writing – original draft, Writing – review and editing; Surojit Sur, Conceptualization, Formal analysis, Supervision, Validation, Investigation, Visualization, Methodology, Writing – original draft, Project administration, Writing – review and editing

## Author ORCIDs

Chetan Bettegowda ⓘ https://orcid.org/0000-0001-9991-7123
Kathleen Gabrielson ⓘ https://orcid.org/0000-0003-3901-4357
Surojit Sur ⓘ https://orcid.org/0000-0003-4536-7343

## Ethics

Six- to eight-week-old female mice were maintained according to the JHU Animal Care and Use Committee.

Reviewer #1 (Public review): https://doi.org/10.7554/eLife.94488.3.sa1

Reviewer #2 (Public review): https://doi.org/10.7554/eLife.94488.3.sa2
Author response https://doi.org/10.7554/eLife.94488.3.sa3

## Additional files

### Supplementary files
Supplementary file 1. Gross pathology and assessment of potential tissue damage in organs of mice treated for 4 weeks (12 doses) with ME3BP-7.

MDAR checklist

### Data availability
Datasets include data from in vitro assays, imaging files from histological slides and ultrasound, and animal study results across multiple figures. Supplementary datasets incorporate publicly available transcriptomic data from CCLE (*Barretina et al., 2012*; *Ghandi et al., 2019*), TCGA-PAAD (*Cancer Genome Atlas Research Network, 2017*), and GTEx. All source data files include original measurements, statistical analyses, and image documentation used in the figures and supplementary materials to reproduce the results are available via Dryad: https://doi.org/10.5061/dryad.vhhmgqp5m.

The following dataset was generated:

| Author(s) | Year | Dataset title | Dataset URL | Database and Identifier |
|---|---|---|---|---|
| Rincon-Torroella J, Dal Molin M, Mog B, Han G, Watson E, Wyhs N, Ishiyama S, Ahmedna T, Minn I, Azad N, Bettegowda C, Papadopoulos N, Kinzler K, Zhou S, Vogelstein B, Gabrielson K, Sur S | 2025 | ME3BP-7 is a targeted cytotoxic agent that rapidly kills pancreatic cancer cells expressing high levels of monocarboxylate transporter MCT1 | https://doi.org/10.5061/dryad.vhhmgqp5m | Dryad Digital Repository, 10.5061/dryad.vhhmgqp5m |

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
