## [Editor Report · eLife assessment]

This study presents a **valuable** finding and developed ME3BP-7 as a novel microencapsulated formulation of 3BP, which specifically targets MCT1-overexpressing PDAC cells. It demonstrates its specificity and efficacy in vitro and in PDAC mouse models, with significant anti-tumor effects and improved serum stability. Overall, the evidence supporting the authors' claims is **solid**.

---

## [Referee Report · Reviewer #1 (Public review)]

Summary:

In this revised manuscript, Rincon-Torroella et al. developed ME3BP-7, a microencapsulated formulation of 3BP, as a potential agent to target MCT1 overexpressing PDACs. The authors provided compelling experimental evidence demonstrating the specific and rapid killing of MCT1 overexpressing PDAC cells in vitro, along with the safety and significant anti-tumor efficacy of ME3BP-7 in multiple PDAC orthotopic mouse models. Overall, this study is very novel, with well-designed experiments and a clear, organized presentation of data that supports the conclusions. The authors have effectively addressed the questions raised in the primary review and provided a thorough discussion of the study's significance, limitations, and future directions, which enhances the readers' understanding of the potential clinical impact of this research.

Strengths:

* Developed a novel agent.

* Well-designed experiments and an organized presentation of data that support the conclusions.

Weaknesses:

No significant weaknesses are noticed.

---

## [Referee Report · Reviewer #2 (Public review)]

Summary:

In the manuscript by Rincon-Torroella et al, the authors evaluated the therapeutic potential of ME3BP-7, a microencapsulated formulation of 3BP which specifically target MCT-1 high tumor cells, in pancreatic cancer models. The authors showed that, compared to 3BP, ME3BP-7 exhibited much enhanced stability in serum. In addition, the authors confirmed the specificity of ME3BP-7 toward MCT-1 high tumor cells and demonstrated the in vivo anti-tumor effect of ME3BP-7 in orthotopic xenograft of human PDAC cell line and PDAC PDX model.

Strengths:

(1) The study convincingly demonstrated the superior stability of ME3BP-7 in serum.

(2) the specificity of ME3BP-7 and 3BP toward MCT-1 high PDAC cells was clearly demonstrated with CRISPR-mediated knockout experiments.

(3) The advantage of ME3BP-7 over 3BP under in vivo situation is highlighted in the revised manuscript.

---

## [Author Response]

The following is the authors’ response to the original reviews.

**Reviewer #1 (Public Review):**
Summary:In the present study, Rincon-Torroella et al. developed ME3BP-7, a microencapsulated formulation of 3BP, as an agent to target MCT1 overexpressing PDACs. They provided evidence showing the specific killing of PDAC cells with MCT1 overexpressing in vitro, along with demonstrating the safety and anti-tumor efficacy of ME3BP-7 in PDAC orthotopic mouse models.Strengths:* Developed a novel agent.* Well-designed experiments and an organized presentation of data that support the conclusions drawn.Weaknesses:There are some minor issues that could enhance the clarity and completeness of the study:(1) Statistical results should be visually presented in Figure 4 and Figure S1.(2) Given the tumor heterogeneity and the identification of focal high expression of MCT1 in Figure 7 and Figure S5B, it is suggested that the authors include the results of immunohistochemical (IHC) analysis of MCT1 expression in both control and ME3BP-7 treated tumor tissues. This addition may offer insight into whether the remaining tumors are composed of PDAC cells with negative MCT1 expression, while the cells with relatively high levels of MCT1 expression were eliminated by ME3BP-7 treatment.(3) The authors are encouraged to discuss the future directions for improving the efficacy of this study. For example, exploring the combination of ME3BP-7 with a glutaminase-1 inhibitor (PMID 37891897) could be a valuable avenue for further research.

We thank the reviewer for pointing these out. We have addressed these individually in detail in the next section

**Reviewer #2 (Public Review):**
Summary:In the manuscript by Rincon-Torroella et al, the authors evaluated the therapeutic potential of ME3BP-7, a microencapsulated formulation of 3BP which specifically targets MCT-1 high tumor cells, in pancreatic cancer models. The authors showed that, compared to 3BP, ME3BP-7 exhibited much-enhanced stability in serum. In addition, the authors confirmed the specificity of ME3BP-7 toward MCT-1 high tumor cells and demonstrated the in vivo anti-tumor effect of ME3BP-7 in orthotopic xenograft of human PDAC cell line and PDAC PDX model.Strengths:(1) The study convincingly demonstrated the superior stability of ME3BP-7 in serum.(2) The specificity of ME3BP-7 and 3BP toward MCT-1 high PDAC cells was clearly demonstrated with CRISPR-mediated knockout experiments.Weaknesses:The advantage of ME3BP-7 over 3BP under an in vivo situation was not fully established.

This is a helpful observation indeed and we have attempted to address this in the revised manuscript as well as clarified the details in the following section in detail.

**Reviewer #1 (Recommendations For The Authors):**
There are some minor issues that could enhance the clarity and completeness of the study:

We appreciate these comments and have addressed them to the best of our abilities in the revised manuscript.

(1) Statistical results should be visually presented in Figure 4 and Figure S1.

Figure 4 and S1 have been updated to include visual representation of statistical results.

(2) Given the tumor heterogeneity and the identification of focal high expression of MCT1 in Figure 7 and Figure S5B, it is suggested that the authors include the results of immunohistochemical (IHC) analysis of MCT1 expression in both control and ME3BP-7 treated tumor tissues. This addition may offer insight into whether the remaining tumors are composed of PDAC cells with negative MCT1 expression, while the cells with relatively high levels of MCT1 expression were eliminated by ME3BP-7 treatment.

This is an excellent suggestion, but unfortunately, we were unable to implement it. We identified a single antibody that showed specificity in our MCT1 knockout isogenic panel after testing 6 different commercial anti-MCT1 antibodies. While the chosen antibody (sc-365501) worked well on fixed human pancreatic cancer samples, it exhibited significant cross-reactivity against background mouse tissue, rendering it difficult to effectively visualize the orthotopically implanted PDx samples.

(3) The authors are encouraged to discuss the future directions for improving the efficacy of this study. For example, exploring the combination of ME3BP-7 with a glutaminase-1 inhibitor (PMID 37891897) could be a valuable avenue for further research.

We have included potentially useful combinations of ME3BP-7 in the discussion section.

**Reviewer #2 (Recommendations For The Authors):**
The overall study is straightforward with translational significance. However, additional clarification is needed to determine the novelty of the study. As cited by the authors, the same group previously published a paper in Clinical Cancer Research, demonstrating the anti-tumor effect of beta-CD-3BP which is also a microencapsulated form of 3BP prepared with succinyl-beta-cyclodextrin. Please clarify what is the major difference between the ME3BP-7 and beta-CD-3BP.

We designed the first generation of beta-CD-3BP and presented the preliminary results in the Clinical Cancer Research paper. Over the last several years, we sought to optimize the formulation so that it would be a a robust clinical candidate. The current manuscript describes our in-depth exploration.

We used a combination of SEC HPLC analyses (representative chromatogram in Fig. 3A) along with a newly developed assay to assess serum stability (representative data in Fig 3B) of a panel of ME-3BP complexes. The panel was created by varying the molar ratios of three different beta-CDs (succinyl beta-CD, native beta-CD and hydroxypropyl beta CD) to 3BP. We discovered that an excess of succinyl-beta-CD (1.2 :1) resulted in the most stable agent with no noticeable batch effects, and this formulation was dubbed ME3BP-7.

The study clearly demonstrated the superior stability of ME3BP-7 in serum compared to 3BP. To further support the advantage of ME3BP-7, it will be important to include the same dose of 3BP as a control in the in vivo treatment experiment to evaluate the difference in both toxicity and anti-tumor effect.

We wanted to include a control arm in our study wherein the same dose of 3BP was used. However, in toxicity studies on three different species of mice, we found that infusion of 3BP at the identical dose was highly toxic, killing the animals within a few days. We have highlighted this toxicity of the non-microencapsulated 3BP in the revised manuscript.